# Barriers and facilitators to use of a mobile HIV care model to re-engage and retain out-of-care people living with HIV in Atlanta, Georgia

**Michelle E. Henkhaus**[1], **Sophia A. Hussen**[1,2], **Devon N. Brown**[1], **Carlos del Rio**[1,2], **Michelle R. Fletcher**[1], **Marxavian D. Jones**[1], **Amulya Marellapudi**[1], **Ameeta S. Kalokhe**[1,2]*

**1** Department of Global Health, Rollins School of Public Health, Emory University, Atlanta, Georgia, United States of America, **2** Division of Infectious Diseases, Department of Medicine, Emory University School of Medicine, Atlanta, Georgia, United States of America

* ameeta.kalokhe@emory.edu

## Abstract

Novel strategies to re-engage and retain people living with HIV (PLWH) who are out of care are greatly needed. While mobile clinics have been used effectively for HIV testing and linkage, evidence guiding their use in providing HIV care domestically has been limited. To guide the development of a mobile HIV clinic (MHC) model as a strategy to re-engage and retain PLWH who are out of care, we aimed to explore stakeholder perceptions of barriers and facilitators to MHC implementation and use. From June 2019-July 2020, we conducted 41 in-depth interviews with HIV clinic providers, administrators, staff, legal authorities, and community advisory board members, PLWH, AIDS service organizations and city officials in Atlanta, Georgia, and domestic and international mobile health clinics to explore barriers and facilitators to use of MHCs. Interviews were transcribed, coded and thematically analysed. Barriers raised include potential for: breach of confidentiality with resulting heightened stigmatization, fractured continuity of care, safety concerns, staffing challenges, and low community acceptance of MHC presence in their locality. Participants provided suggestions regarding appropriate exterior design, location, timing, and co-delivery of non-HIV services that could facilitate MHC acceptance and address the concerns. In identifying key barriers and facilitators to MHC use, this study informs design and implementation of an MHC as a novel strategy for re-engaging and retaining PLWH who are out of care.

## Introduction

Despite widespread availability of potent antiretroviral therapy (ART), less than half the 1.1 million people living with HIV (PLWH) in the United States (US) are virally suppressed [1]. Engaging and retaining patients in HIV care is critical for ART adherence and viral suppression [2], but is limited by several barriers including un-/underinsurance, lack of transportation, housing insecurity, mental health and substance abuse comorbidities, poor social support, inaccessibility of healthcare, social norms, and poverty [3]. In the Southern US,

the Emory IRB (irb@emory.edu) or via the corresponding author.

**Funding:** This study was funded by the National Institutes of Health/National Institute of Allergy and Infectious Diseases through the Emory Center For AIDS Research (P30 AI050409) to ASK. http://www.cfar.emory.edu/; https://www.niaid.nih.gov/; The funders had no role in the study design, data collection, and analysis, decision to publish, or preparation of the manuscript.

**Competing interests:** The authors have declared that no competing interests exist.

medical mistrust [4] and pervasive stigmas (e.g., around HIV [5], minority sexual orientation, poverty, substance abuse, and mental disorders [6]) pose additional retention challenges [7]. Evidence-based retention interventions address many of these barriers and include: provision of social support services, case management [8], child care, housing services, appointment reminders, co-location of services, appointment accompaniment [9], patient navigators [10], peer support [11], educational brochures [12], outreach, and transportation assistance [8, 9, 13–16]. While each of these interventions has demonstrated promise, continued shortcomings in achieving national retention goals suggest the need for new re-engagement and retention strategies tailored to the needs of PLWH who have fallen out of care [17, 18].

Some experts have proposed the use of mobile health clinics to facilitate access to HIV care for difficult-to-reach populations [19]. Mobile health clinics typically deliver health care on a van or bus that can move to different locations and thus provide medical services directly in multiple communities rather than in a traditional, static clinic-based setting. Mobile health clinics have been used for a range of medical purposes (e.g., asthma care, maternal and infant health services) [20, 21]. Within the field of HIV, their use has been largely limited to HIV testing and counselling services [22–27]. Less commonly, mobile health clinics have been used for the provision of HIV care. In sub-Saharan Africa, mobile health clinics are being used to expand HIV services to rural areas in Mozambique [28], Swaziland [29], Kenya [30], South Africa and Uganda [31]. Mobile health clinics are much less common in US-based HIV care, but have been used in the context of needle exchange programs [32] and as part of broader primary care services for people who are homeless [33–35].

Evidence from other settings suggests that mobile health clinics are highly acceptable and have several advantages over traditional clinics [25, 26, 36] including their potential to address transportation barriers, to foster patient-centered care, and to lessen intimidation of patients by traditional healthcare settings [20, 37]. However, there are potential challenges to their use in provision of US-based HIV care that need to be explored prior to implementation. We sought to comprehensively explore key stakeholder perspectives on the barriers and facilitators to the use of mobile HIV clinics (MHCs) as a means of reengaging and retaining PLWH who have fallen out of care in Atlanta, Georgia.

## Methods

### Overview

This qualitative sub-study is nested within a larger mixed-methods pre-implementation study that aims to prepare for utilization of an MHC to re-engage and retain PLWH who are out of care. To explore perceived barriers and facilitators of MHC use, we conducted in-depth interviews with key stakeholders of a large Ryan White-funded infectious diseases clinic, surrounding agencies that provide HIV support services in Atlanta, Georgia, and existing MHCs from June 2019 through July 2020. The study was approved by the Emory University institutional review board and the Grady Research Oversight Committee. Written informed consent was obtained from each subject prior to participation for in-person interviews and oral consent was obtained for interviews conducted by videoconference.

### Participants

Potential participants were purposively sampled to explore a diversity of perspectives about the MHC. Sampling targets included 10 clinic staff and providers, 10 PLWH (5 out-of-care PLWH and 5 PLWH who were clinic Consumer Advisory Board [CAB] members), 10 staff from organizations providing community-based HIV clinical and support services, and 10 institutional and city-based regulatory, billing, and, medico-legal compliance experts [38].

Organizations providing community-based HIV care and support included AIDS Service Organizations (ASOs) working with different groups of PLWH (i.e. gay/bisexual men, women, transgender women, under- and uninsured patients) and mobile clinics providing HIV-related care domestically and internationally. The clinic CAB is an existing group of approximately 10 PLWH who receive HIV care at the study clinic. They were nominated to the CAB based on their interest in HIV advocacy and in enhancing the services provided within the clinic's Ryan White Program. They represent various racial/ethnic, gender and sexual minority groups, and many endorse having had a period in their lives where they were out-of-care or struggled with their HIV diagnosis and appointment adherence and are able to describe strategies they used to overcome the challenges. PLWH who were out of care (i.e. not seen by an HIV care provider in the preceding 6 months) were recruited through the clinic's list of out-of-care patients and from hospital settings. The remainder of participants were recruited through email, direct contact, and the clinic newsletter. Enrollment concluded once the research team determined theoretical saturation was reached.

## Study procedures

Study protocols and interview guides were reviewed by the CAB and revised based on feedback. Interviews were conducted at the clinic or at a mutually convenient off-site private location (i.e. public library, workplace, inpatient hospital room), or by phone or Zoom video conference when participants were unable to meet in person. All interviews were audio-recorded and followed a semi-structured interview guide. Interview guides were tailored to each stakeholder group, and while each guide included open-ended questions about perceived facilitators, barriers, and strategies for using an MHC to deliver HIV care to PLWH who had fallen out of care, probes that followed were tailored to the unique expertise of each stakeholder group. For example, in follow-up to an open-ended question about perceived challenges with mobile HIV care delivery, providers and staff were probed about time, support, and technology challenges, while patients were probed about accessibility and location challenges. New insights from the interviews were shared at weekly research staff meetings and then probed by interviewers in subsequent interviews. The average interview duration was 47 minutes. Participants were compensated $50 for completing the interview.

## Analysis and data management

All interviews were transcribed verbatim. Transcripts were reviewed for quality and de-identified. We commenced coding alongside conducting the interviews, so that inductive codes arising from earlier transcripts could be probed further in subsequent interviews. The codebook was developed after the first three interviews and included deductive codes derived directly from the interview guide and inductive codes that emerged directly from the interviews. To establish consistency of coding and interpretation, definitions and examples of each code were included in the codebook and discussed at research team meetings, all transcripts were coded by two research team members using MAXQDA Plus 20.0.4, and discrepancies in coding were discussed until consensus was reached. When consensus could not easily be reached between the two coders regarding interpretation of a transcript and appropriate code, the code and text in question was presented at the weekly team meeting for further discussion until consensus was achieved. Thematic analysis on coded interviews was applied to examine barriers and facilitators to use of the MHC model.

**Table 1. Participant characteristics.**

| Participant Category | Number | Roles |
|---|---|---|
| People Living With HIV who were out-of-care | 5 | - - |
| Ryan White clinic providers/staff/administrators | 14 | Physician, Nursing staff, Mental Health staff, Social Worker, Peer Navigator, Administrator |
| AIDS-service organization staff | 7 | HIV care, HIV prevention and awareness, primary care inclusive of HIV care |
| Ryan White clinic community advisory board members | 6 | - - |
| Mobile health clinic staff | 5 | County health department, syringe exchange, HIV testing |
| Institutional and city legal and regulatory officials | 4 | Compliance officer, Ryan White program manager, city attorney |
| **Total** | **41** | |

## Results

### Participant characteristics

We conducted 41 interviews (Table 1) with PLWH who were out-of-care (n = 5), Ryan White clinic providers, administrators, and staff (n = 14), ASO staff members (n = 7), CAB members (n = 6), mobile health clinic staff (n = 5), and institutional and city legal and regulatory officials (n = 4). Participants described potential barriers to use of an MHC, that if not addressed would decrease the effectiveness of the model, and also facilitators to enhance the effectiveness of the model and address the identified barriers. Potential barriers and facilitators were organized thematically into patient-level, clinic-level, and environment-level barriers.

### Patient-level barriers and related facilitators

Potential patient-level barriers to an effective MHC model include: concerns around confidentiality and stigma, physical and emotional comfort, patient hesitancy to trust a new clinic with possible new providers, accessibility, the possibility of patients being more likely to fall out of other necessary medical care, and potential for decreased patient responsibility and self-efficacy in care. Facilitators are described in relation to the respective barriers below.

**Confidentiality and stigma concerns.**   Given the persistent stigma associated with HIV, the potential for MHCs to breach patient confidentiality (i.e. through the outside appearance of the van, location, and services provided identifying patients as living with HIV) was a concern brought up by almost every participant, and across stakeholder groups.

> I'd say the challenges and barriers are the stigma of being seen. If, you know, it is—if people recognize this mobile unit as an HIV treatment unit in the community, and then if a member of the community gets on to this mobile unit, people are automatically going to know possibly what this person's HIV status is.
>
> -Mobile health clinic staff

> I mean I feel like it's just more open. Like it's more out there. . .like what, what if somebody who—who wants to know information about why that's there in the community. . . I mean I could see it working for certain communities but umm I think it would be very convenient

in certain, but for me because my family still doesn't know and community and other people I know doesn't know—don't know, so it would be kind of like I wouldn't do it.

-PLWH

Other patients discussed that potential breach of confidentiality was a tolerable risk if it meant receiving accessible care:

Well, I'm comfortable regardless because I know this clinic is something to help me, so I wouldn't be uncomfortable at all. That would be fine. . . because I know that once you go into the clinic, he [the HIV care provider] has to close the door so it's privacy there just between me and him. So the person outside they can't hear what's going on, so I wouldn't feel unsafe.

-PLWH

Participants across all stakeholder groups also described numerous facilitators of patient confidentiality and privacy including incorporation of strategies that diminish chance of disclosure of the true intent of the van (i.e. through nesting HIV care among other services), that detract attention from the MHC in the community, and that limit the likelihood of patients coming across one another on the vehicle.

If you could do like just a general medical bus, where they can get HIV care but also maybe they're doing free checks or health checks, blood pressure checks, you know, um so you're . . .pulling other people in.

-PLWH

I think if it's in a generic area that blends in. Not on the street, but like a strip mall but over to the side where people are not, it's not where you'd just see it. It kind of blends in with what's going on in that neighborhood. . . You know what I mean? People are moving.

-Social Worker

**Lack of physical and emotional comfort.**   Participants also raised concerns regarding patients' physical and emotional comfort, specifically, whether patients would feel welcome, whether they would feel comfortable in the waiting area, and comfortable with the MHC location. These concerns were expressed primarily by PLWH, CAB members, providers and staff in the clinic, and those that worked at ASOs and on mobile health clinics. Facilitators were described by all stakeholders, and included a welcoming outside vehicle appearance, comfortable waiting area with amenities (i.e. water, snacks, air conditioning, and phone chargers), culturally-competent and caring staff with diversity reflective of the community served, and a frequent MHC presence in the community.

Spruce it up with some warm colors and comfortable seats, and . . .it feels better than a waiting room. . . Some people hate hospitals. And just making the care more comfortable, more approachable, less health care-ish.

-ASO staff member

Maybe partner with [a community-based organization that provides meals] and then when they leave out of the clinic then they get a sandwich and something to drink. . .That'll encourage and make them want to come because at least they know when they get there when they leave the doctor they got a sandwich, a dessert, a snack and something to drink, you know.

-PLWH

You know what, have a greeter, have somebody at the door, "Hi, welcome," and stuff. Don't just have us come in and then they sit there and then you say, "yes ma'am." No, greet, "Hi, how are you doing today?" That makes you feel so much more comfortable. It's just the tone of the place, you know that I mean? It's a happy place.

-PLWH

Responding to how one can make an MHC the most effective, acceptable, and safe place as possible, a mobile health clinic staff member described creating familiarity and comfort:

We collaborate with other community-based organizations in the area, so maybe you [the patient] may not be familiar with [our clinic], but you're familiar with another organization that we're collaborating with at the time. Make it more comfortable, and I feel like, you know, more frequent presence in the community, you know, become more familiarized with us. So, that kind of stigma, you know, the stigma is reduced.

-Mobile health clinic staff

**Hesitancy to trust a new clinic and new providers.** Participants, primarily clinic providers and staff and ASO staff members, expressed that patients might be hesitant to trust an MHC model because of the unfamiliarity with receiving HIV care in a mobile setting, possible past negative experiences with HIV testing vans, and potentially new and unfamiliar service providers operating the van.

Maybe just because it's new, right? Just getting people to say, oh, wait, we are doing this in a van? Like, what are we doing? Because the last time somebody pulled up in a van, they were giving me $25 for a survey, right? So just trusting, like, hey, we actually work for [healthcare institution]. . .You can actually have a real appointment here if you want to, right?

-Social Worker

Cons would be, who are these people? Uh does it feel like I'm receiving care from a stranger?

-ASO staff member

However, participants also expressed that trust in the MHC model would be facilitated by communicating the model through providers and peer navigators, word of mouth, patient trust in the larger healthcare institution, provision of quality care, and time itself.

But I think also [healthcare institution] is a trusted brand. I mean, everyone knows the logo, everyone knows the name. It's Atlanta. Like, you're trusted to be like a powerful entity in

the city of Atlanta with regards to healthcare that's able to do things maybe other places aren't. . . [healthcare institution] is a trusted name in healthcare, but also HIV care.

-ASO staff member

I think patients once they see what it can offer, will really put the word out ourselves. So if we make it the best that it can possibly be and really put our best foot forward then our patients will be our best our advocates for it and keep it going.

-Nursing staff

**Accessibility concerns.**   The convenience and accessibility of the van, including location and parking availability, were raised as key factors by clinic staff and providers, ASO staff members, CAB members, and PLWH, impacting the effectiveness of the MHC model.

How are you going to target specific areas if, because Atlanta is so spread out, and I think our population is also kind of spread out. And though there might be some hotspots across Atlanta, I guess, how to be able to reach everyone that might need.

-ASO staff member

We have to find a good, convenient spot that all they got to [do] is if they drive or jump off the bus then they're right there. . . They ain't gonna come if it's a long ways to walk, you see what I'm saying?

-PLWH

**Decreasing patients' responsibility and self-efficacy in their care.**   Bringing HIV care and services to patients might have the unintended effect of reducing patient responsibility and self-efficacy in care, a concern expressed primarily by clinic providers and staff.

Yeah I think it might create this sense for patients um that. . .that the provider is sort of always available to them. It might make them less inclined to um. . .be active in their care. . . The unwitting message that might be sent to the patient is 'you are incapable of coming in right so, so you're fragile so we need to um, we need to make sure you don't leave and that we sort of attend to you in this way.

-Mental health clinician

A second participant expressed concern that patients would fall out of other necessary, less convenient medical care,

Many patients that we see have sort of multiple medical needs, so you know if they sort of need to sort of see a specialist would they be less inclined to then go someplace else to get their [other treatments]?

-Mental health clinician

Some participants suggested that this barrier could be addressed if the MHC could be used successfully as a gateway to re-engagement in long-term care at a traditional fixed clinic.

Yeah, we're gonna be with you for a while, we're gonna get you up on on your feet and then, you need to come to [the regular clinic].

-Healthcare provider

## Clinic-level barriers and related facilitators

Clinic-level barriers include: potential high cost, limited capacity, safety concerns, staffing and time limitations, ability to handle emergencies, adequate awareness of the MHC among the target population, ability to provide HIV care of equal quality to that of traditional clinic-based models, fragmentation of existing patient-provider relationships, and discontinuity of care. Facilitators to address clinic-level barriers are described in relation to the respective barrier.

**Potential high cost.** Some participants, primarily clinic staff and providers, institutional regulatory officials, and ASO staff members, expressed concern over MHC cost-effectiveness, specifically voicing high start-up costs, variable usage, capacity to treat only a few patients at a time, and that it could be a bad steward of providers' time and clinic money in having substantive resources devoted to a few hard-to-reach patients.

Are we looking for ten people to be on the mobile clinic and we only have five today? Is that still a successful thing? So, we have to have that conversation. . .What is the cost benefit factor?

-Compliance official

Other participants, primarily including clinic providers, mobile health clinic staff, and ASO staff, thought it would be cost-effective:

I mean honestly, it seems cheaper, more cost effective than having a brick and mortar.

-ASO staff member

They discussed that scheduled appointments rather than walk-in visits could reduce associated costs.

There could be days where you have no one and it's not a very good um steward of time or money. . .But I think it can be overcome with a schedule.

-Nursing staff

**Limited capacity.** Some, primarily ASO staff members and clinic staff and providers, raised concerns about the limited capacity of an MHC model due to space constraints limiting the number of providers, services provided, and number of patients seen at a given time.

How can you get all of the health care team, all the essential pieces into a little van? I don't know if there would be a model, and this might be thinking way too far outside the box, but having a model instead of just a van, but doing a trailer, dropping off a trailer for, say, a month, and then having more space for these services, and then the next month being in another location and only kind of serving two to three miles radius every month.

-ASO staff member

To address the limited physical capacity of MHCs, participants suggested limiting HIV and non-HIV services to those most critical, integrating with other delivery models like telemedicine, and ensuring strong systems of referral and integration into the larger health care system.

**Safety concerns.** Participants, primarily clinic staff and providers, institutional regulatory officials, ASO staff members, and CAB members, expressed concern for the safety of providers, patients, and supplies, voicing that the MHC could be a target for theft of equipment, medications, and prescription pads, and stigma-related violence if associated with HIV care. Safety concerns were affected by the locations the van operated in as well as community acceptance.

> What would you do about security? Um, there's you know, there's, there's a lot of um hateful people and you know, so you know those are some concerns I have. Um, you know I, I, I'd hate for a mobile clinic—I'd hate for any clinic to be a target of uh of you know any kind of violence or hatred.
>
> -CAB member

To facilitate safety, participants, especially mobile health clinic staff and clinic staff and providers, suggested working with police to determine appropriate locations and area exit plans, notifying police of the schedule, providing staff and providers with safety training, adding security personnel or utilizing the driver as security personnel, limiting the medications on the vehicle, and installing a lockbox, panic button, camera, and signs to facilitate safety.

> The safety thing, it's just a matter of access. Other than people having the misperception that it contains something that it doesn't but that's easily fixed by just saying, . . . "no more than [a certain amount of cash] is on hand anytime" disclaimer out there. There's nothing on here, maybe a computer but yeah, just making its known. No medications, no narcotics, nothing of that nature is here.
>
> -Nursing staff

**Staffing and time limitations.** Participants, primarily clinic staff and providers and institutional regulatory officials, expressed concern with the MHC being staffed by existing clinic staff and providers, who were already being utilized at full capacity. Some, including staff of existing mobile health clinics, suggested staffing the MHC with advanced practice providers to preserve physicians' limited time.

> With my schedule, I can't just go on a mobile clinic. . . would it be like in place of a half-day of clinic, I'd go on the mobile van? Or, am I going to be asked to do it in addition to my clinic slots, which would be very difficult for me to do personally.
>
> -Healthcare provider

**Ability to handle urgent and emergency situations.** To facilitate handling of urgent and emergent situations, participants, primarily clinic staff and providers and institutional regulatory officials, stressed that the MHC model should include appropriate protocols, equipment, and adequate number of personnel.

It's like alright, um. . . somebody falls on the floor, they're half dead, who's gonna help the doctor?

-Nursing staff

They further discussed a need for the MHC to integrate with the health care system to maintain continuity of care during urgent medical situations.

**Adequate awareness of the MHC among the target population.**   Participants across all stakeholder groups raised concerns of inadequate awareness of the MHC among the target population leading to underutilization. They also expressed concern of being able to "market" and communicate about the model to PLWH without targeting patients or exposing it as an HIV-specific van.

There's going to have to be a great deal of effort on the part of the clinic staff to make patients aware and to truly make them aware . . . In my mind I just honestly believe that people should just drive and literally go and locate where people say they are living.

-Nursing staff

Participants noted ways to facilitate awareness: through communication about the MHC option to patients in the fixed clinic before the patients fall out of care (i.e. during clinic orientation), through peer navigators that go into the community to reach patients who have fallen out of care, through flyers in the fixed clinic, through social media, and through word of mouth and visual reminders resulting from the MHC being seen in the community.

Reach people through social media, especially with our current population, so. . . we let the public know where we will be on the mobile unit and when we'll be there.

-Mobile health clinic staff

**Potential lack of HIV care provision of equal quality to the clinic setting.**   Some, including ASO staff members and clinic providers and staff, expressed concern about whether the MHC could provide care of similar quality to a fixed HIV clinic. While some participants voiced that all services provided at the fixed clinic should be available in an MHC model, others discussed that patients should not expect the MHC to provide the same level of care but rather, be used as a re-engagement tool with linkage and integration into the health care system.

The care should be measurable, comparable [to the clinic]. They should be the same, you know what I mean? I, I shouldn't be like, Dang, this? I gotta go in this van, it's broken down, you know? Or I could go to a, the clinic and it's you know. It should, it should be the same. It should be the same look, feel.

-Social Worker

There's a stigma about being poor, and like getting crappy resources for everything. . .the way public health often looks is like look, we're not going to give you anything, except we're going to control this epidemic, right?. . . The danger to me is like, do we further alienate people and you know, when we don't integrate things into a larger care model, like is it

worth it? Like does it maybe have more long-term risks in our ability to like talk about HIV more productively?

-ASO staff member

**Fragmentation of existing patient-provider relationships and discontinuity of care.**
While clinic providers and staff and institutional regulatory officials raised potential fragmentation of existing patient-provider relationships, especially if a patient's primary HIV physician was not on the MHC van, as a concern, they voiced this could be overcome by integration of the MHC into the wider healthcare system.

I love my patients and I love my relationships with them and I don't want to lose it because of, because of this. . . I would want the goal of the HIV um mobile clinic, that that goal is to get you back to me.

-Healthcare provider

## Environmental-level barriers and related facilitators

At the environmental level, barriers to an effective MHC model include community acceptance and safety. Facilitators to community acceptance and safety are also described here.

**Community acceptance and safety.**   Participants across all stakeholder groups except PLWH discussed that the success of the MHC van depended on community acceptance. Negative community perceptions of the MHC van could spur stigmatizing action towards patients, make the van a target for violence or hatred, and inadvertently deter patients from utilizing the services. Participants described that community acceptance could be facilitated by partnering with organizations who are trusted by the community, seeking support from local government, seeking and addressing community concerns before implementation, and embedding HIV care within other health services of benefit to the community's wellness.

If you get a neighborhood up in arms, they're going to come in and meet with their council member and the next thing you know, there's legislation coming out that regulates you.

-City Official

Just reaching out ahead of—before the fact—to the local governments, to the city and the other local governments and finding out you know, developing, establishing the relationships. . .developing partnerships with existing—with existing agencies, is probably a good way to do that, or a good way to get the support of the local governments.

-City Official

## Discussion

By "bringing the clinic to the community," MHCs have potential to revolutionize HIV care for out-of-care PLWH in the U.S. This exploratory study lays critical groundwork for the design of MHC models through examining key stakeholder perceptions about barriers and facilitators to MHC acceptance and use. Factors identified as critical to successful MHC uptake include:

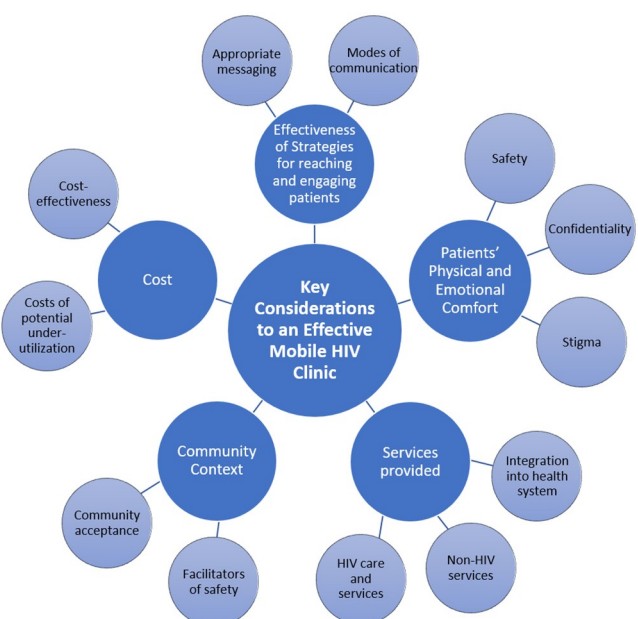

**Fig 1. Key considerations to implementation of a mobile HIV care model to reach people living with HIV who are out of care.**

1) adequacy of strategies to reach and engage PLWH who are out of care, 2) patient sense of physical and emotional comfort inclusive of confidentiality and privacy, 3) breadth of HIV and non-HIV services offered, 4) understanding of the community context in which the MHC is to be implemented, and 5) implementation costs inclusive of opportunity costs (Fig 1).

Common strategies used by other mobile clinics to raise awareness of services among target populations (i.e. vehicle signage, community postings, and television and radio advertising in targeted areas) [27], are not feasible for MHC models given HIV-associated stigma. Instead, the present study and others have highlighted the need for appointments to be scheduled and patients to be notified of the appointments through pagers [39], phone call reminders, or home visits by peer educators. Additionally, MHCs should be located in easily accessible areas (i.e. near communities of patient residence, along public transportation routes) to foster awareness and accessibility. Limited literature guides methods for increasing patient sense of physical and emotional comfort with mobile clinics. Our study affirms strategies used by MHCs in other settings to preserve privacy and confidentiality (i.e. nesting HIV care within other health services, using non-descript MHC labelling), and underscores the importance of establishing safety and emergency response protocols. Study participants suggested equipping the MHC with security personnel, however this needs to be weighed against potential for patient re-traumatization given high baseline rates of incarceration in this patient population. To foster trust, our study emphasized the need to communicate about the MHC model through HIV care providers and peer navigators and to ensure care was provided in a culturally-sensitive manner and of quality equivalent to that provided in the fixed HIV clinic. Ensuring a breadth of HIV and non-HIV services to incentivize utilization, provide comprehensive care, safeguard confidentiality, and mitigate stigma are strategies used by other MHCs and syringe exchange programs [32, 33, 35]. To overcome concerns around limited MHC capacity, care fragmentation, and diminished patient self-efficacy in care and access of other necessary medical care, care provided on the MHC should be integrated into the larger healthcare system (i.e. through strong closed-loop referral systems, incorporation of telemedicine), and could

potentially be used as a transitionary model for patients to ultimately return to the fixed HIV clinic.

Understanding of community context (i.e. safe spaces, parking availability, local regulations, existence of trusted community-based organizations with whom the MHC could partner) was deemed in our study as key to community MHC acceptance, and has been accomplished by others through including community members in the planning [40]. Lastly, other studies of mobile clinics have echoed high start-up and maintenance costs, but deemed them a worthwhile long-term investment [27]. Strategies to increase cost-effectiveness include scheduling appointments, offering services beyond HIV, and using the MHC as a transitionary model to long-term clinic-based HIV care.

The study strengths include its exploration of a new model of HIV care to reach out-of-care PLWH, its examination of a wide breadth of stakeholder perspectives to understand factors affecting MHC adoption, CAB involvement in study conceptualization, and transcript coding by two study team members. Study limitations include recruitment from a single clinic which may affect transferability of findings to other clinics, and the use of a broad definition for "out-of-care" PLWH. The definition unintentionally resulted in inclusion of some patients whose medical appointments were purposefully spaced to greater than 6 months as a result of the patients achieving longstanding viral suppression; thus, some perspectives may not have been reflective of patients who are truly "out-of-care."

In conclusion, this study lays the foundation for the development of a stakeholder-driven MHC to re-engage and retain PLWH. Key factors influencing use of the MHC include the extent to which patients feel physically and emotionally safe (including their perceptions of the potential of the MHC to maintain confidentiality and privacy), the extent to which the MHC provides comprehensive integrated HIV and non-HIV services, the level of community acceptance of the MHC, cost-effectiveness of the model, and the development and use of effective strategies to reach and engage out-of-care PLWH. Next steps include development of the MHC model to address these factors with the continued participation of the various stakeholders, followed by MHC piloting. Future studies will evaluate the effectiveness of the resulting MHC in enhancing retention and viral suppression relative to traditional fixed-clinic HIV care.

## Acknowledgments

We would like to acknowledge the Emory CFAR Prevention Science Core for provision of data collection devices. We express utmost gratitude toward each of the individuals who participated in the study and openly shared their perspectives with us.

## Author Contributions

**Conceptualization:** Sophia A. Hussen, Carlos del Rio, Ameeta S. Kalokhe.

**Data curation:** Michelle E. Henkhaus, Devon N. Brown, Michelle R. Fletcher, Marxavian D. Jones, Amulya Marellapudi.

**Formal analysis:** Michelle E. Henkhaus, Devon N. Brown, Michelle R. Fletcher, Marxavian D. Jones, Amulya Marellapudi.

**Funding acquisition:** Carlos del Rio, Ameeta S. Kalokhe.

**Investigation:** Michelle E. Henkhaus, Devon N. Brown, Michelle R. Fletcher, Marxavian D. Jones.

**Methodology:** Sophia A. Hussen, Ameeta S. Kalokhe.

**Project administration:** Sophia A. Hussen, Ameeta S. Kalokhe.

**Resources:** Ameeta S. Kalokhe.

**Supervision:** Sophia A. Hussen, Ameeta S. Kalokhe.

**Validation:** Michelle E. Henkhaus, Sophia A. Hussen, Devon N. Brown, Michelle R. Fletcher, Marxavian D. Jones, Ameeta S. Kalokhe.

**Visualization:** Michelle E. Henkhaus, Ameeta S. Kalokhe.

**Writing – original draft:** Michelle E. Henkhaus.

**Writing – review & editing:** Michelle E. Henkhaus, Sophia A. Hussen, Devon N. Brown, Michelle R. Fletcher, Marxavian D. Jones, Amulya Marellapudi, Ameeta S. Kalokhe.

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
