## [Decision Letter · Decision Letter 0]

23 Nov 2020

PONE-D-20-30821

Barriers and Facilitators to use of a mobile HIV care model to re-engage and retain out-of-care people living with HIV in Atlanta, Georgia

PLOS ONE

Dear Dr. Kalokhe

Thank you for submitting your manuscript to PLOS ONE. After careful consideration, we feel that it has merit but does not fully meet PLOS ONE’s publication criteria as it currently stands. Therefore, we invite you to submit a revised version of the manuscript that addresses the points raised during the review process.

We look forward to receiving your revised manuscript.

Kind regards,

Bronwyn Myers

Academic Editor

PLOS ONE

Journal Requirements:

2.  Please include additional information regarding the survey or questionnaire used in the study and ensure that you have provided sufficient details that others could replicate the analyses. For instance, if you developed a questionnaire as part of this study and it is not under a copyright more restrictive than CC-BY, please include a copy, in both the original language and English, as Supporting Information, or include a citation if it has been published previously.

3. In the Methods, please discuss whether and how the questionnaire was validated and/or pre-tested. If these did not occur, please provide the rationale for not doing so.

Reviewers' comments:

Reviewer's Responses to Questions

**Comments to the Author**

1. Is the manuscript technically sound, and do the data support the conclusions?

Reviewer #1: Yes

2. Has the statistical analysis been performed appropriately and rigorously? 

Reviewer #1: N/A

3. Have the authors made all data underlying the findings in their manuscript fully available?

Reviewer #1: Yes

4. Is the manuscript presented in an intelligible fashion and written in standard English?

Reviewer #1: Yes

5. Review Comments to the Author

Reviewer #1: This article examined barriers and facilitators to the use of mobile HIV care clinics to re-engage people living with HIV who are out of care. This study fills an important gap by exploring the feasibility of mobile health clinics in the United States. The study strengths include the variety of stakeholders interviewed and engagement of a community advisory board.

There are several areas that need to be strengthened. The methods need to be expanded to describe the sampling and analysis process used. The results need to more clearly indicate which barriers are of most concern to each stakeholder group. Finally, the discussion needs to provide a more a nuanced analysis of the main findings. I offer the following suggestions:

Introduction

Well written!

Methods

• Line 67: Describe the sampling strategy used. Was there a target # for each stakeholder group?

• Line 69: Describe the composition of the Community Advisory Board

• Line 78: Was the same interview guide used for each stakeholder group?

• Line 89 Provide more description of the analysis process. For the line, “...additional inductive codes added based on initial interviews” did coding commence while interviews were being conducted, or does this mean that codes were developed after reviewing a few initial transcripts? Describe the initial process of establishing consistency among coders and the interpretation process. It would be helpful to know how transcripts from each stakeholder group compared.

Results

• The results represent a variety of stakeholder groups which express different barriers and facilitators. For each section, specify which specific stakeholder group is having the concern (e.g. is it only healthcare providers concerned about staffing and time limitations?) to help readers discern which are the most widely held barriers and facilitators for each group.

• Remove Table 2 as it does not add to the results.

• There are currently no quotes representing mobile health clinic staff. Integrate some quotes from this group as they may have the most direct experience with this care delivery model.

• Overall, the results section is quite long. Cut back on the length of some of the longer quotes (i.e. line 318, line 269). Additionally, similar sections should be combined. The sections “potentially falling out of other medical care” and “Decreasing patients’ responsibility and self-efficacy in their care” are similar, as are the sections “community acceptance and safety” and “safety concerns.” Combined into one paragraph.

Discussion

• Line 364: the discussion of all the main findings are condensed in one paragraph. This paragraph should be broken down to add more nuance to findings. For example, the point about increasing security personnel may be in direct contrast to patient concerns about emotional and physical safety. The suggestion in the results that police be brought in may heighten concerns as some PLWH who are out of care may engage in criminalized behaviors. Additionally, there is no discussion about how trust can be built between MHCs and patients.

• Line 389: Add to the limitations that “of the 41 interviews, only 5 were of PLWH.” It is a major limitation that they are underrepresented among study participants.

• Line 393: In the conclusion, include a line or two synthesizing main barriers and facilitators.

6. PLOS authors have the option to publish the peer review history of their article (what does this mean?). If published, this will include your full peer review and any attached files.

Reviewer #1: No

---

## [Author Response · Author response to Decision Letter 0]

22 Dec 2020

Response to Reviewer

Introduction

Well written!

Thank you.

Methods

Line 67: Describe the sampling strategy used. Was there a target # for each stakeholder group?

The study utilized purposive sampling to explore a diversity of stakeholder group perspectives. Our target goal was to achieve 10 clinic staff and providers, 10 people living with HIV (PLWH; 5 PLWH who were out of care, and 5 PLWH who were Consumer Advisory Board members), 10 organizations providing community-based HIV clinical and support services for PLWH (including other mobile clinics, AIDS service organizations, etc.), and 10 institutional and city-based regulatory, billing, and, medico-legal compliance experts. Sample size targets were based on Kvale’s estimate that 15+/-5 individuals from a group is sufficient to reach theoretical saturation.1Although we did not reach the target for the compliance/regulatory officials category, we concluded enrollment once we as a team concluded we had reached theoretical saturation. A description of the sampling strategy has been added to the Methods section on lines 69-72. 

Line 69: Describe the composition of the Community Advisory Board

The Consumer Advisory Board (CAB) was made up of approximately 10 PLWH who receive HIV care at the study clinic. They were nominated to the CAB based on their interest in HIV advocacy and in enhancing the services provided within the clinic’s Ryan White Program. They represent various racial/ethnic, gender and sexual minority groups, and many endorse having had a period in their lives where they were out-of-care or struggled with their HIV diagnosis and appointment adherence and are able to describe strategies they used to overcome the challenges. A description of the composition of the CAB has been added to the Methods section on lines 80-85.

Line 78: Was the same interview guide used for each stakeholder group?

Different interview guides were used for each stakeholder group. While each guide asked open-ended questions about barriers, facilitators, and strategies for using a mobile HIV clinic (MHC) to deliver care, probes that followed were tailored to the stakeholder group. For example, in follow-up to an open-ended question about challenges with mobile HIV care delivery, providers and staff were probed about time, support, and technology challenges in addition to concerns around safety, privacy and confidentiality, while patients were probed about perceived challenges with MHC access and location (in addition to safety, privacy, and confidentiality). Clarification on the differences between interview guides has been added to the Methods section on lines 95-101.

Line 89 Provide more description of the analysis process. For the line, “...additional inductive codes added based on initial interviews” did coding commence while interviews were being conducted, or does this mean that codes were developed after reviewing a few initial transcripts? Describe the initial process of establishing consistency among coders and the interpretation process. It would be helpful to know how transcripts from each stakeholder group compared.

We commenced coding alongside conducting the interviews, so that inductive codes arising from earlier transcripts could be probed further in subsequent interviews. The codebook was developed after the first three interviews and included deductive codes derived directly from the interview guide and inductive codes that emerged directly from the interviews. To establish consistency of coding and interpretation, definitions and examples of each code were included in the codebook and presented and discussed at research team meetings, all transcripts were coded by two research team members, and discrepancies in coding were discussed until consensus was reached. When consensus could not easily be reached between the two coders regarding interpretation of a transcript and appropriate code, the code and text in question was presented at the weekly team meeting for further discussion until consensus was achieved. See below response regarding how transcripts from each stakeholder group compared. Additional description on the analysis process has been added to the Methods section on lines 107-117.

Results

The results represent a variety of stakeholder groups which express different barriers and facilitators. For each section, specify which specific stakeholder group is having the concern (e.g. is it only healthcare providers concerned about staffing and time limitations?) to help readers discern which are the most widely held barriers and facilitators for each group.

For each section within the results, stakeholder groups that primarily brought up each barrier are described. 

Remove Table 2 as it does not add to the results.

Table 2 is now removed.

There are currently no quotes representing mobile health clinic staff. Integrate some quotes from this group as they may have the most direct experience with this care delivery model.

Quotes representing mobile health clinic staff were integrated throughout the results section, within barriers: “confidentiality and stigma concerns” (line 140), “lack of physical and emotional comfort” (line 180), and “Adequate awareness of the MHC among the target population” (line 352). 

Overall, the results section is quite long. Cut back on the length of some of the longer quotes (i.e. line 318, line 269). Additionally, similar sections should be combined. The sections “potentially falling out of other medical care” and “Decreasing patients’ responsibility and self-efficacy in their care” are similar, as are the sections “community acceptance and safety” and “safety concerns.” Combined into one paragraph.

Lines 318 and 269 were shortened. The header “Potentially falling out of other medical care” was removed and the text under this heading was abridged and moved to the header “Decreasing patients’ responsibility and self-efficacy in their care.” We chose to not combine the sections “safety concerns” and “community acceptance and safety" as the themes described under each section are different. The former describes clinic-level safety concerns (i.e. theft of supplies) and informs clinic-level strategies for enhancing safety (i.e. security personnel, installation of panic buttons and cameras, safety training of staff), whereas the latter describes environment-level concerns (including safety concerns) that arise from inadequate engagement of the community (i.e. the neighborhoods where the MHC would be parked) in the process of MHC design and implementation and suggests environment-level strategies to address this concern (i.e. partnering with community-based organizations and local government).

Discussion

Line 364: the discussion of all the main findings are condensed in one paragraph. This paragraph should be broken down to add more nuance to findings. For example, the point about increasing security personnel may be in direct contrast to patient concerns about emotional and physical safety. The suggestion in the results that police be brought in may heighten concerns as some PLWH who are out of care may engage in criminalized behaviors. Additionally, there is no discussion about how trust can be built between MHCs and patients. 

We fully agree with the reviewer and have expanded this paragraph for a more nuanced discussion of the implication of study findings in informing MHC design. Additions are on lines 436-438, 442-447, 450-455.

Line 389: Add to the limitations that “of the 41 interviews, only 5 were of PLWH.” It is a major limitation that they are underrepresented among study participants.

11 of the interviews were PLWH, including 5 who were out of care and 6 from the CAB. Language was added on lines 69-71 of the Methods section describing this sampling strategy. The team unanimously felt that we reached theoretical saturation with the 5 interviews with PLWH who were out of care and 6 PLWH who were on the CAB. Additionally, the qualitative findings presented in this manuscript are part of a mixed-methods study, which includes surveys (not published here) with 50 PLWH who are out-of-care that examine their relative preference for mobile HIV care versus several other community-based models of HIV care delivery and gauge their specific preferences for MHC implementation.

Line 393: In the conclusion, include a line or two synthesizing main barriers and facilitators.

The following line has now been added, “Key factors influencing use of the MHC include the extent to which patients feel physically and emotionally safe (including their perceptions of the potential of the MHC to maintain confidentiality and privacy), the extent to which the MHC provides comprehensive integrated HIV and non-HIV services, the level of community acceptance of the MHC in their locality, cost-effectiveness of the MHC, and the development and use of effective strategies to reach and engage out-of-care PLWH. Next steps include development of the MHC model to address these factors with the continued participation of the various stakeholders, followed by MHC piloting” (lines 474-481).

Additional References

1. Kvale S. Doing Interviews. Thousand Oaks, CA: SAGE Publications; 2008.

---

## [Decision Letter · Decision Letter 1]

5 Feb 2021

Barriers and Facilitators to use of a mobile HIV care model to re-engage and retain out-of-care people living with HIV in Atlanta, Georgia

PONE-D-20-30821R1

Dear Dr. Kalokhe,

We’re pleased to inform you that your manuscript has been judged scientifically suitable for publication and will be formally accepted for publication once it meets all outstanding technical requirements.

Kind regards,

Bronwyn Myers

Academic Editor

PLOS ONE

Additional Editor Comments (optional):

Reviewers' comments:

Reviewer's Responses to Questions

**Comments to the Author**

1. If the authors have adequately addressed your comments raised in a previous round of review and you feel that this manuscript is now acceptable for publication, you may indicate that here to bypass the “Comments to the Author” section, enter your conflict of interest statement in the “Confidential to Editor” section, and submit your "Accept" recommendation.

Reviewer #1: All comments have been addressed

2. Is the manuscript technically sound, and do the data support the conclusions?

Reviewer #1: (No Response)

3. Has the statistical analysis been performed appropriately and rigorously? 

Reviewer #1: (No Response)

4. Have the authors made all data underlying the findings in their manuscript fully available?

Reviewer #1: (No Response)

5. Is the manuscript presented in an intelligible fashion and written in standard English?

Reviewer #1: (No Response)

6. Review Comments to the Author

Reviewer #1: (No Response)

7. PLOS authors have the option to publish the peer review history of their article (what does this mean?). If published, this will include your full peer review and any attached files.

Reviewer #1: No

---

## [Editor Report · Acceptance letter]

18 Feb 2021

PONE-D-20-30821R1 

Barriers and Facilitators to use of a mobile HIV care model to re-engage and retain out-of-care people living with HIV in Atlanta, Georgia 

Dear Dr. Kalokhe:

I'm pleased to inform you that your manuscript has been deemed suitable for publication in PLOS ONE. Congratulations! Your manuscript is now with our production department. 

Kind regards, 

on behalf of

Dr. Bronwyn Myers 

Academic Editor

PLOS ONE